Ontogeny of highly variable ceratitid ammonoids from the Anisian (Middle Triassic)

Bischof Eva Alexandra bischof@uni-bremen.de 1
Schlüter Nils 2
Korn Dieter 2
Lehmann Jens 1
1 Geowissenschaftliche Sammlung, FB5 Geowissenschaften, Universität Bremen , Bremen , Germany
2 Museum für Naturkunde, Leibniz-Institut für Evolutions-und Biodiversitätsforschung , Berlin , Germany
Hedrick Brandon
Electronic publication date: 2021 Mar 2
Publication date: 2021
Volume: 9
Electronic Location ID: e10931
Received 2020 Nov 12; Accepted 2021 Jan 21
Copyright: ©2021 Bischof et al.
Copyright year: 2021
Copyright holder: Bischof et al.
License: This is an open access article distributed under the terms of the Creative Commons Attribution License, which permits unrestricted use, distribution, reproduction and adaptation in any medium and for any purpose provided that it is properly attributed. For attribution, the original author(s), title, publication source (PeerJ) and either DOI or URL of the article must be cited.
License URL: https://creativecommons.org/licenses/by/4.0/

Keywords: Ammonoidea, Ceratitidae, Anisian, Nevada, Geometric morphometrics, Ontogeny, Phenotypic variation, Beyrichitinae, Paraceratitinae, Fossil Hill Member

Funding: German Science Foundation (DFG), project “Nevammonoidea” LE 1241/3-1 This research received support from the German Science Foundation (DFG), project “Nevammonoidea” (LE 1241/3-1). The funders had no role in study design, data collection and analysis, decision to publish, or preparation of the manuscript.

==============================
Ammonoids reached their greatest diversity during the Triassic period. In the early Middle Triassic (Anisian) stage, ammonoid diversity was dominated by representatives of the family Ceratitidae. High taxonomic diversity can, however, be decoupled from their morphologic disparity. Due to its high phenotypic variability, the high diversity of ceratitids of the Anisian of Nevada was initially assumed to be caused by artificial over-splitting. This study aims to contribute data to settle this issue by applying geometric morphometrics methods, using landmarks and semi-landmarks, in the study of ontogenetic cross-sections of ammonoids for the first time. The results reveal that alterations in ontogenetic trajectories, linked to heterochronic processes, lead to the morphologic diversification of the species studied herein. Our knowledge, based on these ontogenetic changes, challenge the traditional treatment of species using solely adult characters for their distinction. This study furthermore demonstrates that the high diversity of the Anisian ammonoid assemblages of Nevada based on the traditional nomenclatoric approach is regarded to be reasonably accurate.

Introduction

After the Permian-Triassic mass extinction event, ammonoids flourished and spread globally to become an important part of the marine biota (House, 1993; Brosse et al., 2013; Brayard & Bucher, 2015; Neige, 2015). They reached their greatest generic diversity of all time in the Triassic Period (Brayard et al., 2009; Whiteside & Ward, 2011). The diversity peak in the late Anisian is dominated by genera of the family Ceratitidae (Brayard et al., 2009; supporting material Fig. S2). Not least due to their wide paleogeographic distribution and high diversity as well as abundance in the fossil record, ammonoids are an excellent biostratigraphic tool. This is especially true for members of the family Ceratitidae Mojsisovics 1879 after which many North American Anisian biostratigraphic zones and subzones are named (Jenks et al., 2015; figs. 13.13, 13.14).

The fossil material used in this study was collected in the late Anisian Fossil Hill Member of the Star Peak Basin in north-western Nevada, USA. The studied successions are considered to be the world’s most complete low-paleolatitude successions, yielding late Anisian ammonoid assemblages (Monnet & Bucher, 2005). The first comprehensive taxonomic work on the Anisian ammonoid communities of the famous fossil locality at Fossil Hill in the Humboldt Range was published by Smith (1914) in his monograph on the North American Middle Triassic marine invertebrates. According to the taxonomic practice of his time, he described or listed a total of 110 ammonoid species from Fossil Hill. More recently, Silberling & Nichols (1982), Bucher (1992) and Monnet & Bucher (2005) refined the original alpha taxonomy and the biostratigraphy with contemporaneous methods and reduced the number to 81 valid species (Brosse et al., 2013). However, it is important to note that succeeding assemblages show a progressive shift in morphology; therefore, the cutoff between contiguous species is essentially arbitrary (Silberling, 1962; Silberling & Nichols, 1982). This challenges the taxonomic concept and sheds new light on diversity patterns in general. An increasing number of studies suggest that the seemingly high diversity could in some cases be artificially inflated by taxonomic over-splitting (i.e., Kennedy & Cobban, 1976; Forey et al., 2004; De Baets, Klug & Monnet, 2013; Knauss & Yacobucci, 2014; De Baets et al., 2015). Furthermore, taxonomic diversity and morphological disparity of Triassic ammonoids were probably decoupled (McGowan, 2004; McGowan, 2005; Brosse et al., 2013). At present, only a few studies have investigated trends in morphological disparity of Triassic ammonoids (Monnet, Brayard & Brosse, 2015).

Previous studies have proven that—particularly due to their accretionary planispiral conch growth with conservation of previous growth stages—ammonoids offer a high-resolution data set for ontogenetic, developmental and also taxonomic studies. While the study of conch ontogeny has a long history in the study of Paleozoic ammonoids (e.g., Korn & Klug, 2007; Korn, 2010; Monnet, De Baets & Klug, 2011; Naglik et al., 2015), it was only rarely examined on Mesozoic ammonoids (e.g., Rieber, 1962; Tajika et al., 2015; Bischof & Lehmann, 2020). So far, the morphology and ontogeny of ammonoids was mainly assessed using descriptive, comparative or traditional morphometric methods (linear measurements). In his classic work of 1966, Raup introduced traditional geometric parameters for the description of coiled conch morphospace. These “Raupian parameters” were subsequently refined by Korn & Klug (2003), Korn & Klug (2007), Korn (2010), Klug et al. (2015a) and Klug et al. (2015b). However, the shapes of discoidal ammonoids often differ through their characteristic ways of ventral arching and presence or absence of a keel. Both characteristics can hardly be described with linear measurements (Neige, 1999). Therefore, the use of traditional morphometric methods might be limited when it comes to distinguishing ceratitid species.

For the first time, the morphology and ontogeny of whorl profiles of the late Anisian ceratitids were analyzed using landmark- and semilandmarks-based geometric morphometric methods (GMM) instead of linear measurements (traditional morphometrics). This is reasoned in the tremendous advantages of GMM over the latter; landmarks and semi-landmarks cover shape variations of complete morphologies, which are sometimes not to be recognized or overseen with linear measurements of traditional morphometrics methods (Neige, 1999). In addition, GMM allow the analysis of shape and size separately (Hammer & Harper, 2005; Zelditch, Swiderski & Sheets, 2012; Polly & Motz, 2016) and do not introduce artifactual patterns of covariation (Gerber, 2017), which is often the case when proportions are studied.

The literature on geometric morphometric analyses (landmark-based approaches and Fourier analysis) of molluscs is rather scarce. Although important pioneering works exist, the previous studies are of limited use in an ontogenetic context because they all focus either on the shape of the whole conch or on single (isolated) ontogenetic stages (landmarks: e.g., Johnston, Tabachnick & Bookstein, 1991; Neige & Dommergues, 1995; Reyment & Kennedy, 1998; Stone, 1998; Neige, 1999; Reyment, 2003; Van Bocxlaer & Schultheiß, 2010; Knauss & Yacobucci, 2014; Fourier analysis: e.g., Courville & Crônier, 2005; Simon, Korn & Koenemann, 2010; Simon, Korn & Koenemann, 2011; Korn & Klug, 2012; Klein & Korn, 2014).

In order to evaluate the hitherto used taxonomic scheme, ontogenetic patterns within the family Ceratitidae and their changes over time were investigated. Working with ontogenetic cross-sections allows the estimation of the relative age of the whorls, which adds an extra dimension to the analysis. The tools presented here are intended to complement traditional descriptions and to evaluate and quantify their results. This study should serve as a general motivation to conduct GMM studies on invertebrates with accretionary planispiral growth.

Materials & Methods

Geological setting

The ammonoid material derives from the Fossil Hill Member of Fossil Hill in the Humboldt Range and Muller Canyon in the Augusta Mountains (Pershing County), north-western Nevada, USA (Fig. 1) and is stored in the Geosciences Collection of the University of Bremen (GSUB), Germany. The material from the Wilderness Study Area of the Augusta Mountains, Pershing County was collected with permission of the US Department of the Interior, Bureau of Land Management (BLM, Nevada State office, Winnemucca District). The Fossil Hill Member is a succession of alternating layers of mudstone with lenticular limestone and calcareous siltstone beds (see Fig. 2). The rich and diverse fossil content consists primarily of halobiid bivalves and ammonoids. Detailed geological and stratigraphic descriptions were published by Nichols & Silberling (1977), Silberling & Nichols (1982) and Monnet & Bucher (2005).

Figure 1 Location of the study area in NW Nevada, USA. The Fossil Hill and the Muller Canyon localities are marked. Figure adapted from Bischof & Lehmann (2020), Fig. 1.

Figure 2 Biostratigraphic distribution of fossil material and synoptic lithostratigraphic sections of the outcrops in the Muller Canyon and Fossil Hill area. Stratigraphic section of Muller Canyon adapted from Bischof & Lehmann (2020), Fig. 2.

Gray areas in stratigraphic column: Calcareous siltstone; white areas: lenticular limestone, box width refers to weathering profile.

Studied specimens

The fossil material comprises 72 ammonoid specimens of the family of Ceratitidae (Mojsisovics, 1879). These represent twelve species in seven genera (Fig. 2, Table 1) that either belong to the subfamily Beyrichitinae (Spath, 1934) or Paraceratitinae (Silberling, 1962). Most of the studied species show high intraspecific variation with overlapping morphologies (see Table 1 and Figs. 3–5). Members of these genera (Gymnotoceras, Frechites and Parafrechites in particular) are sometimes hard to differentiate. They mainly differ in the ventral conch outline, ornamentation, adult ribbing and maximum growth size. The younger the individuals are, the greater the similarities. Despite their complicated taxonomy, all selected species are index fossils of the late Anisian Fossil Hill Member (see Fig. 2). It was assumed that the individual species have similar coiling rates (i.e., the individual species develop the same number of whorls in the course of their life). The total number of volutions developed by the species varies between five and a half and seven (see Table 1—Total number of volutions).

Table 1 Morphological comparison of the species in focus. For biostratigraphic distribution see Fig. 2.

Species	N	Total number of volutions	Venter and conch outline	Sculpture	Dmax [mm]	U/D	W/D	Figure herein	
Beyrichitinae Spath, 1934									
Billingsites cordeyiMonnet & Bucher, 2005	6	6—6.5	Slightly angular ventral shoulder	Falcoid, prorsiradiate ribs, sometimes branched	34.3	min: 0.17	min: 0.28
max: 0.35	3A–D	
			Very weak developed keel	Nodes at branching points		max: 0.24			
Dixieceras lawsoni (Smith, 1914)	10	6—7	Stout, discoidal outline	Falcoid, prorsiradiate ribs, sometimes branched	57.7	min: 0.19	min: 0.23
max: 0.44	3I–L	
			Rounded ventral shoulders	Umbilical thickening of whorls		max: 0.25			
Frechites nevadanus (Mojsisovics, 1888)	6	5.5—6	Subrectangular outline	Strong, falcoid, prorsiradiate ribs, sometimes branched	28.9	min: 0.29	min: 0.39
max: 0.46	3M–P	
			Clearly developed keel	Adults: Pronounced tubercles at lower flank		max: 0.37			
Frechites occidentalis (Smith, 1914)	7	6—7	Angular ventral shoulder	Strong, slightly prorsiradiate ribs, some rare tubercles	42.6	min: 0.24	min: 0.38
max: 0.43	4I–L	
			Sometimes very weak developed keel	Towards maturity ribbing fades		max: 0.27			
Gymnotoceras blakei (Gabb, 1864)	5	5.5—6	Discoidal outline	Falcoid, prorsiradiate, unbranched ribs	37.8	min: 0.15	min: 0.30
max: 0.38	4A–D	
			Rounded ventral shoulders, weak keel	Towards maturity fading ribs and megastriae		max: 0.28			
Gymnotoceras mimetusMonnet & Bucher, 2005	9	6—6.5	Discoidal to subrectangular outline	Megastriae and weak falcoid, prorsiradiate ribs, slightly swelling towards umbilicus	43.0	min: 0.14	min: 0.29
max: 0.40	5K–N	
			Rounded ventral shoulders, no keel			max: 0.22			
Gymnotoceras rotelliformis (Meek, 1877)	6	6	Stout discoidal outline, very weak keel	Regular, slightly prorsiradiate ribs	34.3	min: 0.17	min: 0.32
max: 0.38	4M–P	
			Rounded ventral shoulders	Towards maturity ribbing slightly fades		max: 0.26			
Gymnotoceras weitschatiMonnet & Bucher, 2005	3	6	Compressed, discoidal outline	Megastriae and weak falcoid, prorsiradiate ribs, slightly swelling towards umbilicus	28.4	min: 0.17	min: 0.29
max: 0.33	5A–E	
			Perfectly rounded shoulders, no keel			max: 0.20			
Parafrechites dunni (Smith, 1914)	5	5.5—6.5	Stout discoidal outline, sometimes keel	Regular but weak, slightly prorsiradiate ribs	35.2	min: 0.18	min: 0.31
max: 0.42	4E–H	
			Rounded to subangular ventral shoulders	Towards maturity ribbing slightly fades		max: 0.20			
Parafrechites meeki (Mojsisovics, 1888)	5	5.5—6	Subrectangular outline	Strong and regular, falcoid, prorsiradiate ribs, sometimes branched	32.1	min: 0.22	min: 0.34
max: 0.41	5O–R	
			Strong keel, sub-angular shoulders			max: 0.27			
Paraceratitinae Silberling, 1962									
Brackites vogdesi (Smith, 1904)	4	6—7	Subrectangular outline, slightly rounded shoulders	Regular, falcoid, branched, prorsiradiate ribs	29.6	min: 0.28	min: 0.35
max: 0.37	3E–H	
			Tubercles at brancing point		max: 0.37			
Marcouxites spinifer (Smith, 1914)	6	5.5—6	Subrectangular outline, angular shoulder	Strong and regular, falcoid, prorsiradiate ribs	25.8	min: 0.26	min: 0.38
max: 0.42	5F–J	
			Clearly developed keel	Tubercles and spines at branching point		max: 0.36			
Notes.

N Number of specimens

U maximum umbilical diameter

W maximum whorl width

D maximum diameter of conch

Measurement values and ratios based on material herein. More detailed information on the studied species was published by Silberling & Nichols (1982) and Monnet & Bucher (2005).

Figure 3 Ceratitid ammonoids from the Anisian (Middle Triassic) Fossil Hill Member of NW Nevada, USA.

(A–D) Billingsites cordeyi (Monnet & Bucher, 2005), (A, B) GSUB C11082, (C, D) GSUB C11517; (E–H) Brackites vogdesi (Smith, 1904), (E, F) GSUB C11649, (G, H) GSUB C11646; (I–L) Dixieceras lawsoni (Smith, 1914), (I, J) GSUB C13801, (K, L) GSUB C13805; (M–P) Frechites nevadanus (Mojsisovics, 1888), (M, N) GSUB C12377, (O, P) GSUB C12382.

Figure 4 Ceratitid ammonoids from the Anisian (Middle Triassic) Fossil Hill Member of NW Nevada, USA.

(A–D) Gymnotoceras blakei (Gabb, 1864), (A, B) GSUB C12243, (C, D) GSUB C12264; (E–H) Parafrechites dunni (Smith, 1914), (E, F) GSUB C9946 (G, H) GSUB C12906; (I–L) Frechites occidentalis (Smith, 1914), (I, J) GSUB C8998, (K, L) GSUB C13251; (M–P) Gymnotoceras rotelliformis (Meek, 1877), (M, N) GSUB C11594, (O, P) GSUB C11702.

Figure 5 Ceratitid ammonoids from the Anisian (Middle Triassic) Fossil Hill Member of NW Nevada, USA.

(A–E) Gymnotoceras weitschati (Monnet & Bucher, 2005), (A, B) GSUB C11111, (C–E) GSUB C11158; (F–J) Marcouxites spinifer (Smith, 1914), (F, G) GSUB C10050, (H–J) GSUB C10137; (K–N) Gymnotoceras mimetus (Monnet & Bucher, 2005), (K, L) GSUB C15005, (M, N) GSUB C13811; (O–R) Parafrechites meeki (Mojsisovics, 1888), (O, P) GSUB C12534, (Q, R) GSUB C12618.

Preparation and data acquisition

We prepared high-precision cross-sections intersecting the protoconch of each specimen, following the methods by Korn (2010), Klug et al. (2015a) and Klug et al. (2015b). Subsequently, we scanned the polished surfaces in high resolution with a flat screen scanner to ensure that all pictures have the same scale. Thereafter, the scan images were digitized. CT scan images of Anisian ammonoids from Nevada do not provide sufficient contrast of the internal structures for a reliable analysis (Bischof & Lehmann, 2020).

Based on the digitized cross-sections, we performed a 2D landmark-based geometric morphometrics analysis. The landmarks were retrieved in tpsDig2 v.2.31 (Rohlf, 2010). Sixteen landmarks were digitized per half whorl (i.e., whorl stage), which resulted in 176 landmarks per specimen (16 landmarks on 11 half whorls; Fig. 6). This set of landmarks consists of two single (1, 2) and 7 pairs of landmarks (3–16), of which eight are sliding semi-landmarks. Whereas landmarks are discrete anatomical loci (i.e., point of highest curvature of venter), sliding semi-landmarks are placed along a curve (or a surface) between two landmarks in a way that best describes the curvatures of the outline. In a second step, an algorithmic approach optimizes the approximation of the outline (Zelditch, Swiderski & Sheets, 2012).

Figure 6 Digitized sketch of high-precision cross-section of an ammonoid specimen meeting the initial chamber (protoconch) with position of landmarks on last two half whorls. Filled crosses: fixed landmarks; empty crosses: sliding landmarks; black numbers: numbers.

Definition of fixed landmarks: (1) venter of preceding whorl; (2) venter of whorl; (3) and (4) ventral shoulder or point of highest curvature; (5) and (6) maximum width; (7) and (8) Umbilical seam.

In order to omit missing values in subsequent analyses, the data set was limited to whorl stage number 5.5. From a methodological point of view, it is more practical to rotate the shells by 90° compared to conventional illustrations (cf. Stridsberg, 1990) into a lying position. Since ammonoid conchs are spiral-shaped, each whorl is cut in two parts when preparing the cross sections (see Fig. 6). The half whorls on the left side of the protoconch have odd numbers (“odd whorls”; here 0.5–5.5) and those on the right side have even numbers (“even whorls”; here 1.0–5.0”). Homologous landmarks were set in accordance to the axial plane.

Calculation of procrustes shape

All geometric morphometric analyses were carried out using the R software v 3.6.3. (R Core Team, 2020) packages Morpho v2.8 (Schlager, 2017), geomorph v3.3.1. (Adams et al., 2020) and RRPP v0.6.0 (Collyer & Adams, 2018; Collyer & Adams, 2020). Plots were drawn with the R package ggplot2 (Wickham, 2016). Using the Morpho::procSym function, the 2D landmark coordinates were subjected to a full generalized Procrustes alignment (GPA). The semilandmarks were slid minimizing Procrustes distance. The full Procrustes fit standardizes size, orientation and position, leaving only the Procrustes shape coordinates (Bookstein, 1991, chap. 7.1, p. 258–270; Hammer & Harper, 2005; Zelditch, Swiderski & Sheets, 2012). Since the “odd” and “even” whorls cannot be made congruent by any of these operations (i.e., alignment, translation, rotation), all “even” whorls were manually mirrored before the GPA.

The individual whorls were regarded as different structures of the ammonoid conch. Therefore, the GPA was performed separately for every whorl. The procSym function performs Procrustes superimposition including sliding of semi-landmarks on curves and accounts for the symmetry of the object. Subsequently, the R function geomorph::combine.subsets was used to normalize the configurations of all whorl stages to unit centroid size or with a customized weighting (see “Developmental morphospaces”). The centroid size (CS) is regarded as a proxy for the size of the whorls and equals the square root of the summed squared distances of each landmark from the centroid of the landmark configuration before the GPA (Zelditch, Swiderski & Sheets, 2012). The function geomorph::combine.subsets was originally introduced to combine different parts of a body (e.g., heads and tails; Collyer, Davis & Adams, 2020).

To visualize the multivariate data in two-dimensional morphospaces, we ran a Principal Component Analysis (PCA) on the aligned Procrustes shape coordinates using the R function stats::prcomp. Thereby we used two different types of visualization: Ontogenetic trajectory spaces and developmental morphospaces.

Ontogenetic trajectory spaces

It is well-known that ammonoids have a very characteristic but also complex ontogenetic development (e.g., Klug, 2001). To visualize the ontogenetic development of ammonoids, there are different types of morphospaces. Ontogenetic trajectory spaces (originally called ontogenetic morphospaces), as defined by Bischof & Lehmann (2020), p.2), illustrate the differences in total ontogenetic development of individuals. They show the data in an artificial state of combined morphologies of different ontogenetic stages. To calculate an ontogenetic trajectory space, all Procrustes shapes (i.e., whorls) of an individual are re-assembled before running the Principal Component Analysis. This means that, in an ontogenetic trajectory space, the ontogenetic trajectory of every individual is reduced to a single data point. Ontogenetic trajectory spaces are a tool to examine if the ontogenetic pathways of individuals differ, but they do not show how the trajectories vary. To test whether ontogenetic trajectories statistically differ between species, a multivariate analysis of variance (MANOVA) using the R function stats::manova was applied.

Developmental morphospaces

Developmental morphospaces as defined by Eble (2003, p. 40) are morphospaces that directly contain developmental information. In terms of this study, this means that every individual dot in the morphospace reflects a specific ontogenetic stage (i.e., half whorl) of an individual. By connecting all points of an individual, its ontogenetic trajectory can be obtained. In contrast to ontogenetic trajectory spaces, developmental morphospaces show how individual whorls differ from each other.

General Procrustes Analysis (GPA) removes all information about size from a given set of data leaving only the pure shape coordinates. However, as can be seen in Fig. 6, size differences between different whorl stages are tremendous. If normalized to unit centroid size (i.e., non-weighted morphospace), the earliest whorls of ammonoids therefore get enormously enlarged and the last whorls scaled down. In general, deviations (measurement uncertainties as well as actual morphological variation) are increased for the initial whorls and reduced for older whorl stages. Therefore, a second morphospace with weighted Procrustes shape coordinates was calculated. Thereby, the logarithmic centroid size (log10CS) of all configurations of a whorl stage were normalized to the proportional centroid size of the respective stage to the sum of all whorl stages (log10 CSwhorl i/∑ log10 CSwhorls). The principal components of the PCA on the weighted shapes were called wPC (weighted principal components).

If the relative log10CS is used to normalize the centroid size of the configurations, this approach is extremely similar to a Relative Warp Principal Component Analysis (RW-PCA; size-shape space) after Mitteroecker et al. (2004). To calculate a RW-PCA the shape matrix of a configuration is augmented by an additional column containing information about the log10CS of the configurations. Whereas the R function geomorph::combine.subsets scales every configuration accordingly, the size information in the RW-shape matrices are stored in the additional variable. The resulting RW size-shape space can be analyzed with an ordinary PCA. Typically, RW size-shapes spaces are strongly dominated by the log10CS and PC1 therefore often accounts for more than 90% of the variation. If proportional log10CS-values (log10CSconfiguration i/∑log10CSconfigurations) are used, the analysis is less dominated by size, but the eigenvalues are very similar to the ones of the weighted PCA (wPCA). For simplicity the R function was used here.

Because weighting does not change the shapes itself, weighted and non-weighted developmental morphospaces look very similar. The main difference is the placement of the individual configurations within the morphospace. Whether weighted or non-weighted shape coordinates should be used, depends on what the analysis is intended to show. A summary of the three different approaches how to combine landmark configurations can be found in Table 2.

Table 2 Comparison of most important characteristics of weighted and non-weighted approaches to combine landmark configurations.

Shape variables	Effect on shape data set	Advantages	Disadvantages	
Non-weighted (wPCA)	• Minimizes shape difference between all configurations of all o.s.	• No perturbation of original dataset	• Overestimation of deviations of earliest/small earliest/smallest o.s. and underestimation of latest/largest o.s.	
Weighted (wPCA)	• Adds allometric/size information to analysis	• More complicated computation than other methods	• Domination of size (less than in RW-PCA)	
	• Maximizes shape differences between o.s.	• Deviations of earliest/smallest o.s. less overestimated and of latest/large o.s. less underestimated	• Partial loss of objectivity	
	• Minimizes shape differences within an o.s.		• May suppress potential true variation in earliest o.s.	
			• Slight perturbation of original data set (creation of arbitrary covariances)	
Shape-size space (RW-PCA)	• Adds allometric/size information to analysis	• Easy to compute	• Extreme domination of size	
	•Every configuration is scaled individually according to their centroid size	• Intuitive	• Strong perturbation of original data set, cannot be used for most subsequent analyses	
Notes.

o.s. ontogenetic stages (i.e., whorls)

To model the shapes at the maximum and minimum PC-values, the R function GeometricMorphometricsMix::reversePCA (Fruciano, 2019) was used. The function is designed to recalculate artificial Procrustes shape variables from the extreme PC-values in a morphospace. The thin-plate spline deformation grids were calculated using the R function geomorph::plotRefToTarget.

Trajectory analysis

In morphometric studies, ontogenetic trajectories represent a series of measurement values of different ontogenetic stages of an individual or a group, called longitudinal data (Klingenberg, 1998). To quantify the differences of the ontogenetic trajectories of the individual species, the R function RRPP::trajectory.analysis with 999 iterations was used. The function calculates a linear model with at least one categorical interaction variable (here: Shape ∼ Species * WhorlStage) and assesses differences in path distance (magnitude differences, length of trajectories), trajectory shape and the angle between the individual trajectories (trajectory correlation) (Collyer & Adams, 2013).

If weighted shape-coordinates were used, the artificial size-shape relationship could overlay true differences between the trajectories. Therefore, only non-weighted shape coordinates were analyzed in the trajectory analysis.

Results

Ontogenetic trajectory spaces

Ontogenetic trajectory spaces are means to visualize whether or not the ontogenetic development of two or more individuals differs. The first three components (PCs) of the Principal Component Analysis (PCA) on the shape coordinates with combined ontogenetic stages of an individual account for 58.3% (PC1 = 38.3%, PC2 = 13.4%, PC3 = 6.6%) of the total variation. Especially considering that there are a total of 352 primary components (x and y coordinates of 176 landmarks), but only 72 specimens, this result can be regarded as satisfactory. The convex hulls of the ontogenetic trajectory space of most species reveal a large overlap (Fig. 7). Nevertheless, ontogenetic trajectory spaces differ significantly between species (MANOVA: Pillai’s trace = 9.8962; ∼ F (1, 11) = 1.6437, p < 0.001).

Figure 7 Ontogenetic morphospace of all species analyzed. (A) Principal Component 1 and 2; (B) Principal component 1 and 3.

Since PC1 accounts for 38.3% of the total variation, the most important characteristic is the position of the individuals on the x-axis. In fact, there are certain species that primarily have negative PC1 values (B. vogdesi, F. nevadanus, F. occidentalis, M. spinifer, P. meeki) and some that are more restricted to positive PC1 values (G. blakei, G. mimetus, G. rotelliformis, G. weitschati, P. dunni). B. cordeyi and D. lawsoni, both cover a wider range of different PC1 values, but are generally restricted to negative PC2 and positive PC3 values.

Non-weighted developmental morphospace occupation

The first three components of the PCA on the non-weighted shape space account for 93.8% (PC1 = 78.5%, PC2 = 11.5%, PC3 = 3.1%) of the total variation. The PCA plot (Fig. 8) of PC1 and PC2 shows that the whorls of early ontogenetic stages cover the lower left quadrant of the morphospace (negative PC1 and PC2 values), which characterizes extremely depressed, broad whorls with a flat venter (Fig. 9A). The center of the morphospace (PC1 equals 0 and PC2 is positive) is occupied by intermediate growth stages (juveniles), which have a more quadratic outline with an only slightly triangular venter (Fig. 9B). The lower right quadrant (high PC1 and low PC2 values) is associated with the latest ontogenetic stages (adults). Towards maturity, the whorls increase mainly in height and have a clearly triangular venter and sometimes a keel (Figs. 9C and 10). Overall, there are two extreme adult shapes: Type (A) describes rather depressed, stout conchs with only a slight overlap with the preceding whorl and are associated with much shorter ontogenetic trajectories. Type (B) describes compressed conchs with a clearly triangular venter and a higher degree of overlap and are associated with longer ontogenetic trajectories. For the assignment of the species to the two types see Table 3.

Figure 8 Developmental morphospace with PCA of Procrustes shape variables. Point size refers to whorl stage. Deformation grids of the mean shape to the modeled shapes of the extreme values for PC1 and PC2.

Figure 9 Mean shapes of whorl stages 0.5, 3.0 and 5.5.

Fixed landmarks are marked with a cross.

Figure 10 TPS spline of mean shape of whorl 5.5 of all species in this study (grey) plotted against the mean shape of whorl 5.5 of the respective species (black).

Fixed landmarks are marked with a cross.

The developmental morphospace of beyrichitine and paraceratitine ammonoids comprises three basic shape stages, which are not separated by sharp borders (Fig. 9): (1) Earliest whorls: broad and very flat; (2) Juveniles: more rounded and depressed; (3) Adults: mostly high and compressed whorls. Since type A species stop their development at more rounded and depressed whorls, their adult whorls resemble the juvenile stages of type B (Fig. 10).

Table 3 Summary and explanation on the three different ontogenetic types. Heterochronic terms as defined by McNamara (2012).

Type	Species	Heterochrony	Adult whorl shapes	
A1	F. nevadanus	Paedomorph	Depressed, stout conches, only slight overlap with preceding whorl	
F. occidentalis	
M. spinifer	
B. vogdesi	
P. meeki	
B	B. cordeyi	Peramorph (Acceleration)	Compressed conches, more pronounced venter, more overlap with preceding whorl	
D. lawsoni	
G. blakei	
G. mimetus	
G. rotelliformis	
G. weitschati	
P. dunni	

Ontogenetic trajectories in the non-weighted developmental morphospace

The ontogenetic trajectories of species in the non-weighted developmental morphospace share many similarities: They all have similar directions of propagation and a slight parabolic shape (Fig. 8). The variation detected by the trajectory analysis (R function RRPP::trajectory.analysis) revealed significant differences in trajectory length (path distance), trajectory shape and trajectory slope between most species (File S1, summarized results below).

Members of the type A ontogeny have smaller magnitudes of shape change and different trajectory shapes than members from type B ontogeny (path distances A: 0.2005–0.0225; B: 0.2276–0.2778). For the assignment of the species to the two types see Table 3. Only pairwise differences of the path distance and the trajectory shape between type A and type B species are statistically significant (magnitude of shape change: 17/66 possible pairs; trajectory shape: 16/66 possible pairs).

Most species have statistically significant pairwise differences in trajectory slope (57/66 possible pairs). Species of all pairs with non-significant pairwise p-values are in the same ontogenetic group (i.e., both belong to type A or B). The trajectories of the nine pairs with non-significant differences in slope, have non-significant magnitudes and shapes also. However, none of the species that share a common slope have overlapping biostratigraphic ranges.

Weighted developmental morphospace occupation

The first three components of the PCA on the weighted shape space account for 94.8% (wPC1 = 85.8%, wPC2 = 7%, wPC3 = 3.0%) of the total variation. In comparison to the regular PCA, the wPCA is—by definition—more strongly controlled by the centroid size of the configurations, which is mainly expressed by the domination of PC1.

Similar to the regular PCA plot (Fig. 8), the wPCA morphospace (Fig. 11) can be divided into three main parts: (1) The extremely depressed earliest whorls cover the lower left quadrant (low PC1 and PC2 values); (2) the center of the plot (PC1 equals 0, PC2 positive) is occupied by the more depressed whorls of juveniles and (3) adult whorls are associated with positive PC1 values. In contrast to the PCA, the wPCA reveals a more distinct separation of the type A and type B groups of adult whorls (see Table 3). Representatives of the more depressed type B clearly occupy the lower right quadrant (positive PC1 and negative PC2 values). This division into type A and B can also be seen in the mean shapes of the whorl 5.5 of the respective species (Fig. 10).

Figure 11 Developmental morphospace with PCA of weighted Procrustes shape variables. Point size refers to number of whorl stage. Deformation grids of the mean shape to the modeled shapes of the extreme values for PC1 and PC2.

Discussion

Members of the family Ceratitidae show high intraspecific variation and strongly overlapping morphospaces (Table 1, Figs. 3, 4 and 5). The ornamentation, which is often regarded as essential for the description of Mesozoic ammonoid groups (Klug et al., 2015; Klug et al., 2015b), is not a unique characteristic among the family Ceratitidae. A better feature to delineate the ceratitids studied here appears to be the shape of the whorl section. The latter, however, cannot be quantified adequately by traditional morphometric methods (Neige, 1999). Accordingly, the utility of conventional taxonomic and morphological methods is limited in this regard. Here, we utilize landmarks and semi-landmarks on ontogenetic cross-sections. Since previous geometric morphometric studies on mollusks all focus either on conch shape or on single (isolated) ontogenetic stages (landmarks: e.g., Johnston, Tabachnick & Bookstein, 1991; Neige & Dommergues, 1995; Reyment & Kennedy, 1998; Stone, 1998; Neige, 1999; Reyment, 2003; Van Bocxlaer & Schultheiß, 2010; Knauss & Yacobucci, 2014; Fourier analysis: e.g., Courville & Crônier, 2005; Simon, Korn & Koenemann, 2010; Simon, Korn & Koenemann, 2011; Korn & Klug, 2012; Klein & Korn, 2014) they cannot be regarded as ontogenetic studies. This study, investigates the use of geometric morphometric methods (GMM) with respect to their usefulness in ontogenetic developmental studies and taxonomic descriptions.

Ontogenetic patterns in Ceratitidae

The ontogenetic trajectories of the studied species comprise the biphasic development from strongly depressed to weakly depressed to compressed whorl profiles (Figs. 8–11). It is commonly accepted that sudden changes in ontogenetic allometry often mark the onset of sexual maturity (i.e., Kullmann & Scheuch, 1970; Klug, 2001; Klug et al., 2015a; Klug et al., 2015b).

The studied species can be divided into two main ontogenetic groups: Type (A) Truncated trajectories that are associated with depressed adult whorls; type (B) longer, complete trajectories that lead to a compressed adult whorl shape. The process of lengthening and shortening of the trajectories (i.e., related to changes in rate and timing of the development) account for the ontogenetic differentiation of the species in focus. This contrasts a previous traditional morphometric analysis by Bischof & Lehmann (2020) of ptychitids, which revealed that the spherocone-cadicone morphospace is much more distinct. The highly ontogenetically differentiated genus Ptychites directly differed through characteristic ontogenetic trajectories.

While precise temporal growth rates of ammonoids are unknown (Lécuyer & Bucher, 2006; Knauss & Yacobucci, 2014), a basic assumption herein was that the individual species have similar coiling rates (i.e., the individual species develop the same number of whorls in the course of their life). Modified rate/timing of shape change from any ancestor to any descendent within an evolutionary framework is called heterochrony (Zelditch, Swiderski & Sheets, 2012, p. 317). Between type A and type B species, interspecific variation of the species in focus arises from an acceleration, a special case of peramorphosis; (for discussion of this term, see Alberch et al., 1979; McNamara, 2012) that allows type B species to occupy an extended portion of the morphospace characterized by more compressed whorls. Therefore, the studied ceratitids do not primarily differ in shape, but rather in the timing of the development of individual shapes. Heterochrony as a mechanism in macroevolution is known to be a key driving factor in phenotypic diversification (e.g., Gould, 1977; Alberch et al., 1979; McKinney & McNamara, 1991; Gerber, Neige & Eble, 2007; Gerber, 2011; Korn et al., 2013; Knauss & Yacobucci, 2014). The quantification of patterns of morphologic disparity and the relationship between size and shape (i.e., heterochrony) will be the subject of future studies.

Anisian ammonoid diversity

It is widely agreed that ammonoid diversity reached its maximum during the Triassic period (House, 1993; Brayard et al., 2009; Whiteside & Ward, 2011). Thereby, the late Anisian ammonoid diversity peak was dominated by members of the family Ceratitidae (Brayard et al., 2009; supporting material Fig. S2). However, there are a growing number of studies critically questioning diversity peaks by arguing that—to some extent—the high diversity might be artificially inflated by taxonomic over-splitting (Forey et al., 2004; De Baets, Klug & Monnet, 2013; Knauss & Yacobucci, 2014).

The results obtained here do justice to the general opinion that ontogenetic trajectories can be a powerful tool to describe (e.g., Korn & Klug, 2007) and discriminate ammonoid species (e.g., Rieber, 1962; Bischof & Lehmann, 2020): The newly introduced methods succeeded in statistically discriminating the ontogenetic pathways of the pre-defined ceratitid species. Based on this analysis, the high diversity of the Anisian ammonoid assemblages of Nevada appears not to be artificially inflated and the alpha taxonomy is regarded to be adequate. However, the high morphological resemblances of the investigated species cannot be denied. Therefore, this study supports the main idea of McGowan (2004), McGowan (2005) and Brosse et al. (2013) that taxonomic diversity and morphological disparity need not necessarily be closely linked.

It is important to be aware of the fact that GMM carry no direct biological information. They help to understand if and how configurations differ, but not what the underlying mechanisms for their morphological development are. In the complex discoidal morphospace landmark-based approaches have proven to be useful to evaluate a priori defined taxonomic groups. Nevertheless, geometric morphometric methods cannot be considered as being a phylogenetic or taxonomic tool per se. But they certainly represent an improvement and valuable supplement to traditional methods.

Why it is worth the effort

There is no doubt that preparation and analysis of ontogenetic cross sections involves a lot of work (Korn, 2012). However, geometric morphometric methods (GMMs) open the door to a new world of objectified, statistically quantifiable descriptions. For example, in the case of the fauna described herein, conventional descriptions and traditional morphometric methods did not succeed to differentiate species adequately. Landmarks and semi-landmarks, however, make it possible to statistically quantify shape variations of entire morphologies (Neige, 1999) and allow the analysis of shape and size separately (Hammer & Harper, 2005).

The high resolution of the ontogenetic trajectories of the herein studied material was achieved owing to the accretionary planispiral growth of ammonoids with conservation of previous growth stages (Korn, 2012), which adds an intuitive, relative time-component to the ontogenetic analysis. Even though it is likely that small-scale ontogenetic changes are overlooked at a measurement density of one measurement per 180 degrees, it can be assumed that no major developmental steps were skipped (Tajika & Klug, 2020). Leaving out complete ontogenetic stages would most likely prevent the recognition of ontogenetic processes such as heterochrony. If, for example, only the earliest and latest stages of the ceratitid development were analyzed, representatives of type A and type B would differ fundamentally. The accretionary growth of many ammonoid conchs therefore not only adds an individual time component to the analysis, but more importantly ensures that no major developmental steps have been overlooked. This reinforces the general opinion that ontogenetic trajectories of ammonoids are a powerful tool to study evolutionary processes.

Conclusions

The Anisian ammonoid diversity peak was dominated by the family Ceratitidae (Brayard et al., 2009; supporting material Fig. S2). However, the investigated ceratitid species show high intraspecific variation and sometimes completely overlapping morphospaces. Using conventional methods, ceratitids are often difficult to distinguish. It was therefore assumed that the high Anisian diversity in Nevada might be artificially inflated by taxonomic over-splitting.

Using a landmark-based geometric morphometric approach, this study succeeded in differentiating the ontogenetic growth of the pre-defined taxonomic entities in the fossil material from the late Anisian Fossil Hill Member in Nevada, USA. Based on the findings of this study, the high Anisian ammonoid diversity in western North America appears not to be unreasonably inflated. In this context, this study furthermore supports the hypothesis that taxonomic diversity and morphologic disparity of Triassic ammonoids were decoupled (Brosse et al., 2013; McGowan, 2004; McGowan, 2005). The largest interspecific differences of ceratitids are the result of alterations of the ontogenetic trajectories that are likely linked to heterochronic processes (i.e., differences in timing of ontogenetic changes). This means that the individual species of this group are not solely defined by the morphology they attain at a certain growth stage, but rather by the sum and timing of all of their ontogenetic stages. The statistical quantification of the relationship between size and shape (i.e., heterochrony) will be the subject of future studies. These processes make an ad hoc distinction of the different species particularly challenging.

For a reliable traditional taxonomic identification of the species herein, it is necessary to have several individuals (Silberling, 1962) with different ages of the same species from the same stratum. It has furthermore proven to be essential to analyze morphological variation of ceratitids not only between species but also across different ontogenetic stages. Therefore, the significance of ontogenetic studies on ammonoids with regard to taxonomic implications cannot be dismissed. The geometric morphometric methods introduced herein represent a big leap towards more quantitative and objective taxonomic descriptions of ammonoids.

Supplemental Information

File S1 Results of trajectory analysis using the R function RRPP:trajectory.analysis.

Table S1. Absolute Path distances of ontogenetic trajectories.

Table S2. Pairwise comparison of magnitude of shape change (path distance). Green highlighted cells indicate a p < 0.05.

Table S3. Pairwise comparison of magnitude of trajectory shape. Green highlighted cells indicate a p < 0.05.

Table S4. Pairwise results trajectory correlation (angle between trajectories). Green highlighted cells indicate an p < 0.05.

Click here for additional data file.

Data S1 Scripts and data

Click here for additional data file.

We would like to thank D Kuhlmann (now Messel, Germany) as former technician of the Geowissenschaftliche Sammlung Bremen for mechanical preparation of the material collected by our working group. M Krogmann (Bremen, Germany) is thanked for his broad support in the artwork for this article. We are indebted to K Boos for statistical support and the introduction to the R software. P Embree (Orangevale, CA, USA) is thanked for broad support, including organization of field-campaigns, scientific information and the permission to collect on his private property. Our gratitude goes to the reviewers Christian Klug and Kenneth De Baets and the editor Brandon Hedrick for their corrections and suggestions that greatly improved the manuscript.

Additional Information and Declarations

Competing Interests

Author Contributions

Field Study Permissions

Data Availability

The authors declare there are no competing interests.

Eva Alexandra Bischof conceived and designed the experiments, performed the experiments, analyzed the data, prepared figures and/or tables, authored or reviewed drafts of the paper, preparation of fossil specimens, and approved the final draft.

Nils Schlüter conceived and designed the experiments, performed the experiments, analyzed the data, authored or reviewed drafts of the paper, and approved the final draft.

Dieter Korn conceived and designed the experiments, authored or reviewed drafts of the paper, preparation of fossil specimens, and approved the final draft.

Jens Lehmann conceived and designed the experiments, authored or reviewed drafts of the paper, and approved the final draft.

The following information was supplied relating to field study approvals (i.e., approving body and any reference numbers):

The U.S. Department of the Interior, Bureau of Land Management (BLM, Nevada State office, Winnemucca District) gave permission to collect samples in the Wilderness Study Area of the Augusta Mountains, Pershing County.

The following information was supplied regarding data availability:

Data and code are available in the Supplemental Files.

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
