# Peer review of "Ontogeny of highly variable ceratitid ammonoids from the Anisian (Middle Triassic)"

_PeerJ, doi:10.7717/peerj.10931_

## Round 0.1 · original submission · Minor Revisions

Dear authors,

Thank you for submitting your manuscript. Based on my reading and reviewer comments, I would be happy to recommend this for publication in PeerJ pending revisions. For your resubmission, please include a clean version of the manuscript, a tracked changes version showing all changes that were made, and a response to reviewers document outlining how you responded to all reviewer comments. Please see my more specific comments below:

Best,

Brandon P. Hedrick, Ph.D.



I would like to see more of a comparison with previous work using Fourier methods. I do not work on ammonoids, but I had been under the impression that a substantial literature exists examining evolution in ammonoids using Fourier analysis. You mention this in your introduction (lines 95–100), but it does not return in the discussion. An additional paragraph in your ‘why you should take the trouble’ section comparing GM with Fourier analysis in the study of ammonoid evolution would be valuable.

I am surprised at the lack of discussion of phylogenetic comparative methods. Although I feel that phylogenetic PCA can be problematic (e.g. Uyeda et al., 2015), it would definitely be useful to assess the impact of phylogeny on your shape data. This can be done relatively easily in geomorph using the physignal function.

Additionally on lines 275–280 in the results, you discuss some comparisons of ontogenetic trajectories. I am not sure what you did and this must be made more clear in the methods. If this used comparisons across taxa, which I think it did, then you need to justify why you did not use pgls analysis or other phylogenetic comparative methods (or use them).

Make sure references are in chronological order in the text. I noticed this on line 333–34 and line 338, but check throughout.

Line 44: ‘which are the namesake for’

Line 63: ‘only a few’

Line 72: Which of these papers developed the Raupian parameters? If they all did, perhaps cite them at the end of the sentence. As it is here, it is difficult to read.

Line 77: ‘are limited’

Line 79–100. These three paragraphs have redundancies and could be combined and shortened.

Line 105: Evaluate rather than ‘objectify’

Line 121: ‘These represent 12..’

Line 140: ‘resulted’

Line 156: Hammer and Harper (2005) seems like a strange citation to me for this. I would cite either Bookstein’s original work or more commonly, Zelditch et al. (2012), which is the second edition of Zelditch et al. (2004), pre-dating the 2005 paper. Also line 360.

Line 161: Were semi-landmarks slid by minimizing bending energy or Procrustes distance?

Line 201: This is more commonly referred to as ‘size-shape space’

Line 235: ‘means’

Line 238: Principal components analysis is capitalized here, but not in the methods. It doesn’t matter which, but be consistent.

Line 243: What do you mean by satisfactory here? Be specific

Line 251: No need to write out principal components analysis here. Just PCA is fine since it has been previously defined. Also line 287.

Line 347: ‘discriminating’

Line 351: comma rather than semicolon

Line 368: You might consider calling the time component a ‘growth component’. Since ammonoids are so commonly used for biostratigraphy, I feel like ‘time component’ suggests geologic time rather than time in growth.

·

Basic reporting

The English is very good. I pointed out some minor problems in the pdf.
I suggested one additional paper to cite.
The structure is clear, methods well-explained and state-of-the-art.
The content is well intelligible.
Interpretations are backed up by data.

Experimental design

-original, i.e. this was not carried out in the same way before with ammonoid data
-research question is well-defined
-thoroughly gathered data
-methods well described in great detail

Validity of the findings

The ms does address an interesting topic. It has been suspected that diversity is artificially inflated in the fossil record (ammonoids and other groups). This hypothesis was tested and falsified for a Triassic ammonoid family here.
From a method and statistical point of view, I have no complaints.

Additional comments

The main thing you could improve is writing out the taxon names in figures 7,8, 10 and 11. There is always enough space and the reader immediately knows where the data is from.

·

Basic reporting

The manuscript is scientifically written and clear and unambiguous, professional English is used. Most appropriate literature is cited with some rare exceptions (see annotated PDF).

Experimental design

The methods are appropriate – I would like to see a larger discussion/comparison on the use of weighted or unweighted approaches (this could be done by producing a table). Also as far as I understood you did pairwise comparisons of trajectories in developmental morphospace but I guess this could also be supplemented by doing pairwise comparisons of variance of individual species within the ontogenetic "trajectory" morphospace. Also the names are a bit confusing as developmental more or less is a synonym of ontogenetic (maybe use ontogenetic trajectory space versus developmental space).

Validity of the findings

The interpretations are appropriate. The underlying data and code are provided and robust. Their name should however be corrected as they are currently addressed as "Field study approval documentation" (already in use for the actual field study approval documentation).

Additional comments

I greatly enjoyed reading your paper and seeing such novel analyses. In additional to those remarks, i have 3 additional points
1) Some relevant references are missing (see annotated pdf)
2) It would be good to differentiate in most graphs the landmarks and semi-landmarks (e.g., Fig. 9-10) for clarity and completeness sake.
3) A more philosophical point: are continuous differences in ontogenetic development sufficient to separate species? This is rather an interpretation which might be appropriate but not necessarily true. Who is to say that time-averaged assemblages belonging to the same species within beds which likely did not overlap during their lifetimes where not exposed to different environmental conditions during their ontogenies and therefore look morphologically different/had differences in ontogenetic development. I would therefore suggest to express yourself therefore slightly more carefully that “traditionally defined species also differ in ontogenetic trajectories which might support the idea that might be different species”.

These and additional suggestions can be found in the annotated pdf.

---

## Round 0.2 · Minor Revisions

Dear authors,

Thank you for your careful attention to reviewer comments in the previous round of reviews. I am happy to move this paper forward to the next stage. However, there are a few additional grammar issues that should be dealt with first (see below). Once these are corrected, I believe the manuscript will be ready for acceptance.

Best,

Brandon Hedrick, Ph.D.


Line 35: ‘time’

Line 40–41: Grammar issue. Perhaps, ‘This is especially true of members of the family Ceratitidae, for which many North American Anisian biostratigraphic zones and subzones are named’

Line 83–84: Grammar: ‘Perhaps allow the analysis of shape and size separately’. I am not sure what you mean by ‘easier to use for statistical analyses’. Delete that part or clarify.

Line 98: ‘allows the estimation of’

Line 169: ‘in two-dimensional morphospaces’

Line 184: add comma, ‘differ between species, a multivariate’

Line 238: ‘means to visualize whether or not the ontogenetic…’

Line 242: ‘there are a total of’

Line 259: ‘a more quadratic’

Line 319: Not sure what you mean by ‘content’ here. Reword.

Line 339: ‘and’

Line 378: ‘allow the analysis of shape and size separately’

Line 401: ‘succeeded in differentiating’

---

## Round 0.3 · accepted · Accept

Dear authors,

Thank you for your submission and your careful consideration of reviewer comments. I think that this paper is now ready to be accepted in PeerJ. I noted a few final grammatical errors that need to be fixed prior to publication:

Line 40: space between ‘1879 for’

Line 65: ‘set’ instead of ‘basis’

Line 350: ‘there are a growing number’

Please let me know if you have any questions. I look forward to seeing the paper published.

Best,

Brandon P. Hedrick, Ph.D.